# Mapping the Sensory Fingerprint of Swedish Beer Market through Text and Data Mining and Multivariate Strategies

Gonzalo Garrido-Bañuelos [1,*], Helia de Barros Alves [1] and Mihaela Mihnea [2,3,*]

1   Agriculture and Food, Product Design—RISE—Research Institutes of Sweden, 41276 Göteborg, Sweden; heliadba@gmail.com
2   Material and Exterior Design, Perception—RISE—Research Institutes of Sweden, 41276 Göteborg, Sweden
3   School of Hospitality, Culinary Arts and Meal Science, Örebro University, 71260 Grythyttan, Sweden
*   Correspondence: gonzalo.garrido.banuelos@ri.se (G.G.-B.); mihaela.mihnea@oru.se (M.M.)

**Abstract:** The continuous increase of online data with consumers' and experts' reviews and preferences is a potential tool for sensory characterization. The present work aims to overview the Swedish beer market and understand the sensory fingerprint of Swedish beers based on text data extracted from the Swedish alcohol retail monopoly (Systembolaget) website. Different multivariate strategies such as heatmaps, correspondence analysis and hierarchical cluster analysis were used to understand the sensory space of the different beer styles. Additionally, sensory space for specific hop cultivars was also investigated. Results highlighted Gothenburg as the main producing area in Sweden. The style Indian Pale Ale (IPA) is the largest available at the retail monopoly. From a sensory perspective, commonalities and differences were found between beer types and styles. Based on the aroma description, different types of ale and lager can cluster together (such as Porter and Stout and Dark lagers). Additionally, an associative relationship between specific aromas and hop cultivars from text data information was successfully achieved.

**Keywords:** beer; sensory science; text data; data analysis; market study



## 1. Introduction

Beer has been part of our culture for centuries, witnessing society's evolution from ancient to modern times [1]. It has also been a significant source of research, with studies varying from understanding its raw materials, such as malt [2] or hops [3–5], to investigating its impact on health [6]. Numerous studies have also investigated beer flavor chemistry, stability and sensory properties [1,7–10]. Beer perception is undoubtedly influenced by its chemical composition [11–13], but also by cultural factors [14,15] or the level of expertise of the taster [16].

Bear flavor terminology was developed during the late 1970s. Meilgaards and colleagues developed the beer flavor wheel [17] to provide an unambiguous terminology that could help communication in quality control and product development. About a decade ago, Schmelzle [18] decided to update and improve this terminology because some of its olfactory, gustatory, and haptic sensory perceptions were found to overlap, and some terms were not adequately matched with specific sensory perceptions. This updated beer aroma wheel is currently used as an essential tool for selecting terms in descriptive and profiling sensory tests. The flavor spectrum of terms includes 72 attributes grouped into seven areas, which describe the six primary odor and aroma classes (fruity, floral, vegetal, spicy, heat-induced, and biological) and taste and texture [18]. The characterization of the sensory space of traditional and novel beer products has conventionally been achieved through the use of different sensory methodologies [19–21] and the level of expertise (experts and trained or untrained consumers) [22–26].

However, the continuous increase of online data has become a reliable source to understand consumers' preferences and a potential tool for sensory characterization [15,27–30].

For example, De Brito et al. used data extracted from "Untappd", a social network specialized in beer, to understand the cultural differences in beer preferences. The results showed that cities belonging to the same country varied slightly in preference for beer compared to cities in other countries. Similarly, Valente et al. used text data extracted from John Platter's *Wine Guide to South African Wines* guide to investigate the sensory space of varietal wines (Chenin blanc and Sauvignon blanc), modelling its sensory space [28]. Their work also led to the update and re-evaluation of the previous Chenin blanc aroma wheel (developed in 2005) based on Platter's data. [28,31]. Similarly, Da Silva et al. [29] used text mining strategies to generate an aroma wheel for apple juices. Mafata investigated different data fusion strategies to understand the relationship between chemical and sensory markers in wine [32]. Different statistical methods have also been explored in beers [33]. The most recent work is being performed in craft gins and beers [27]. In this case, Kruger and colleagues combine data mining with an automated procedure for attribute consolidation based on lexicon standardization (based on gin wheel and beer aroma maps).

The present work aims to map the Swedish beer market and its sensory profile through the information provided by Systembolaget [34]. Systembolaget is the governmental Swedish alcohol retail monopoly established in the mid-1800s to minimize alcohol-related problems by responsibly selling alcohol without a motive to profit. Besides the information provided by the product producer, for every product found in Systembolaget, additional information, such as sensory description, bitterness, or fullness, is provided. Therefore, the aim of the present work is to highlight the potential applicability of using text data in sensory science by, firstly, giving an overview of the current Swedish beer market, secondly, applying different data analysis and visualization strategies to map the sensory space of Swedish beer styles, and finally, by exploring the usability of text data as a possible tool to understand the sensory characteristics of specific ingredients, such as different hop cultivars.

## 2. Materials and Methods

### 2.1. Data Capture, Standardisation, and Data Mining

Data were manually extracted between 9 March 2020 and 3 April 2020 from the website of Systembolaget for the categories "Öl" and "Sverige" (respectively "beer" and "Sweden" in Swedish). Figure 1 illustrates a workflow with the different steps pursued during this study. The study focused on "ale" and "lager", and therefore, "sour beers" were not included. Data were captured in Swedish and posteriorly translated to English. The following data was captured: name of the beer, beer type, beer style, producer, supplier, producing area, sensory description. Also, hop-related information was included when available.

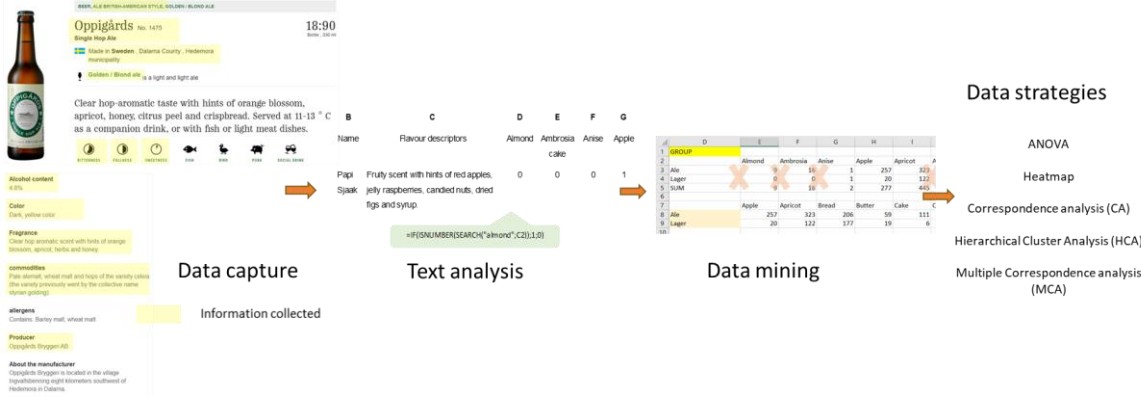

**Figure 1.** Flowchart illustrating the workflow process from data capturing to multivariate analysis.

The attributes were tabulated as variables in the columns. Function words, prepositions, articles, or hedonic terms were excluded. Synonyms and attributes expressed in both singular and plural forms were merged.

### 2.2. Data Analysis

Demographic distribution of beers/breweries was performed with the function Maps from Excel (Excel Office 365—Version 2009, Microsoft Corp., USA). Pie charts were used to create a visual overview of the type of beers. One-way ANOVA on price (SEK/L) per region was performed with software STATISTICA (Statistica 13.2 (TIBCO Statistica software, Palo Alto, CA, USA)). Fisher's least significant difference (LSD) corrections were used for post hoc analyses. Significant differences were judged on a 95% significance level ($p < 0.05$). The sensory characterization was performed with different multivariate strategies using XLStat 2021.3.1 (Addinsoft (2021), New York, NY, USA). Contingency tables were built with sensory attributes as variables and beers as rows. Firstly, characterization of the sensory space of the main beer types (ale and lager) was approached with Heatmap analysis, including hierarchical cluster analysis (HCA) for both rows and columns. Secondly, correspondence analysis (CA) and subsequent HCA were performed separately on ale and lager datasets. The aim was to understand the sensory space of the beer styles within these two beer groups. Lastly, multiple correspondence analyses were used to explore the aroma profile of specific hop cultivars.

### 3. Results and Discussion

#### 3.1. General Overview of the Swedish Market

In March 2020, a total of 1910 Swedish ale and lager beers were registered in Systembolaget. Most of the beer production is found in the three main urban areas: Västra Götaland län (Gothenburg area), the leading producer (Figure 2A), Stockholm län (Stockholms), and Skåne län (Malmö). This beer distribution agrees with the number of breweries/microbreweries, the Västra Götaland being the "beer" region by excellence in Sweden (Figure 2B). ANOVA results showed specific differences in price (Figure S1). The diversity of beer styles led to a high error bar. However, there is an interesting result to highlight as is average, Gothenburg beers are significantly more expensive than those from Stockholm (Figure S1).

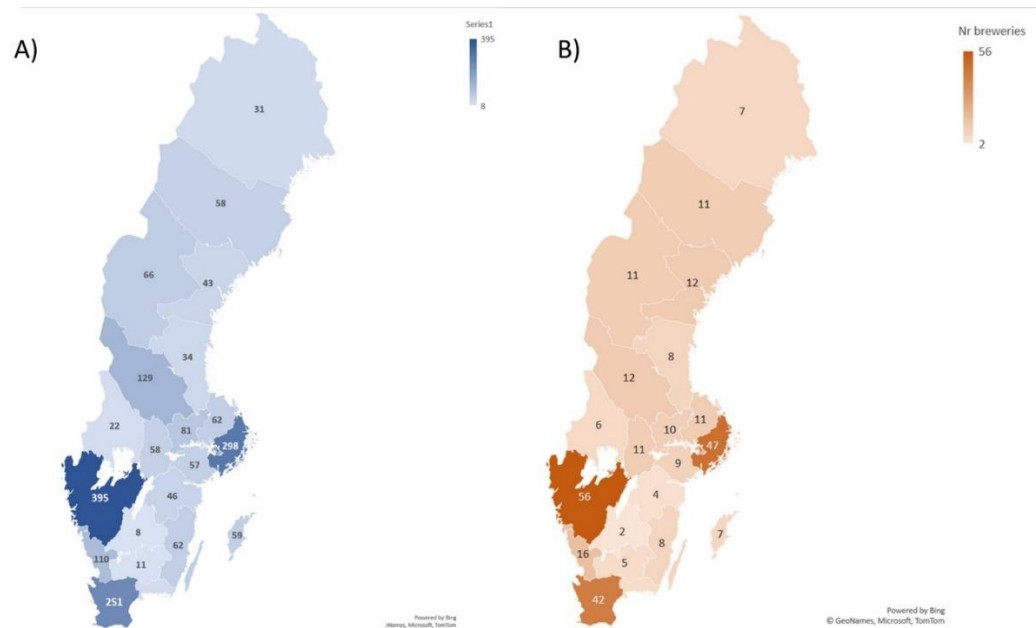

**Figure 2.** (**A**) Demographic distribution of Swedish beers available at Systembolaget in March 2020. (**B**) Distribution of breweries in Sweden.

Out of the analyzed 1910 beers, 1322 were classified as "ales", whereas 588 were "lagers". This considerably larger amount of "ale" beers available at Systembolaget can be considered an indicator of the Swedish consumer's preferences. Beers were subclassified based on the beer types and styles. Therefore, for ales, we found Belgian Ale, British-American Ale, Porter and Stout, and German type. Thirteen beers were described as "ale" but did not have a specific classification. These beers were excluded from the sensory characterization. British-American ales were found to represent 75% of the ale section (Figure 3A). Within British-American ales, the beer style Indian Pale Ale (IPA) accounted for the largest number, with 443 bears (Figure S2A). For lager beers, we found light lagers, medium dark lagers, and dark lagers (Figure 3B), light lagers being the largest group. On average, the price per liter of the three types of lagers is lower than that of any bear in the ale groups, especially Porter and Stout beers. The price range varies from 21.8 SEK/liter for light lager to 305.33 SEK/liter for a beer from Porter and Stout's group (data not shown).

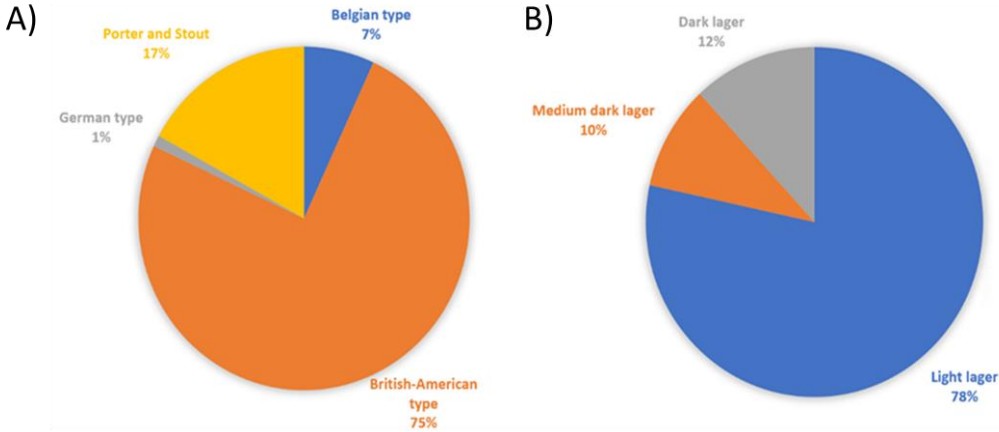

**Figure 3.** Pie-chart illustrating the representative % of the different types of "ale" (**A**) and "lager" beers (**B**).

### 3.2. Sensory Fingerprint of Swedish Beers

#### 3.2.1. Lexicon of Swedish Beers

A total of 116 different sensory attributes (from 294) were obtained after the standardization and data mining process. The number of sensory descriptors was more extensive for ale beers (111) than lager (23). The style with the richest lexicon was for "British-American" beer style, for which 106 different attributes were used. Within Swedish lagers, Swedish "light lager" beers were the more complex (based on their vocabulary assortment) with 60 different descriptors. This is partly due to the specific brewing processes and types of yeast used for ale and lager beers [35], which directly affect the formation of specific aroma molecules such as specific esters producing fruity aroma [36]. Additionally, the use of specific ingredients, for instance, hops containing aroma molecules such as terpenes [37,38] and thiols [12,39], could contribute to these aromatic descriptors. It is essential to mention that not all beers contained hops. The use of hops was cited 1583 times (63% being unknown, i.e., the hop cultivar is not specified) for ales, whereas only 535 times (75% unknown) for lagers. On many occasions, multiple hop cultivars are used during the beer process, increasing this citation number. This could also be reflected in a sensory point of view: when comparing the five most used attributes to describe "ale" beers, we found "orange", "malty", "hoppy", "fruity", and "dried fruit", whereas for "lager" beers we found "malty", "bread", "honey", "orange", "herb(s)". The term "hoppy" was used as a descriptor in 27.3% of Swedish "ale" beers, whereas it was only used in 2.0 % of Swedish lagers. More detailed information about the hops will be discussed in the following section. The type of malt also has a vital role in the flavor profile, as described by Parker [1]. However, this parameter was not included in this study.

### 3.2.2. Sensory Characterization

The first approach towards understanding and characterizing the sensory space of Swedish beers was using a heatmap on the main beer styles. To reduce noise, this analysis was exclusively performed on attributes with a frequency of citation >50, similarly to the approach taken by Valente [31]. In heatmaps, the data matrix's rows (beer types) and columns (sensory attributes) are clustered independently using ascendant hierarchical clustering based on Euclidian distances. Both rows and columns are then permuted according to the corresponding clustering, bringing similar columns (sensory attributes) closer to each other and similar lines (beers) closer to each other. This creates a graph reflecting data in the permuted matrix where data values are replaced by color intensities). This technique has already been successfully applied in wines to investigate wine quality drivers for Chenin blanc and Pinotage wines [40].

Therefore, in Figure 4 we can observe how beer styles from ale and lager clustered on a mixed form. The first cluster is composed of two sub-clusters, grouped "Belgian ale", "British-American", "German style ales", "light lager", and "medium lager". Attributes such as "fruity", "apple", "grapefruit", "bread", "honey", "malty", "orange", "dried fruit", "syrup", "spice", and "apricot" were highly used to describe subcluster of "Belgian ales" and "British American ales". The advantage of using heatmaps is that differences within the same cluster can immediately be visualized. For example, the terms "hop", "apple", or "pineapple" were frequently used in British-American beers but not in "Belgian ales". (Figure 4). The second subcluster grouped "medium lager", "German type ale", and "light lager" with the following descriptors: "apricot", "fruity", "bread", "honey", "crispbread", "herb", "malty", "orange", and "syrup".

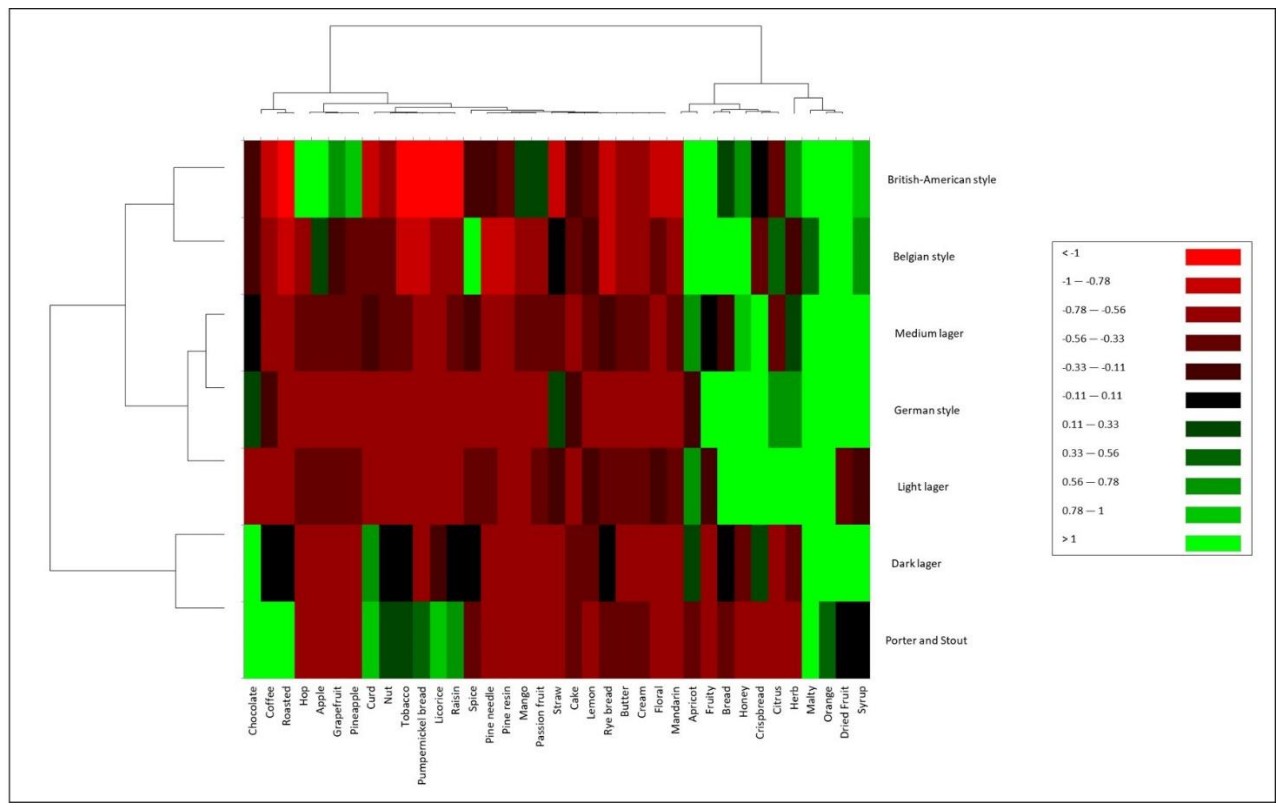

**Figure 4.** Heatmap illustrating the sensory attributes and beer styles. The dendrogram on the left corresponds to the hierarchical cluster analysis (HCA).

Finally, the second clustered grouped of "dark lager" and "porter and stout", is described with attributes such as "chocolate", "coffee", "roasted", "curd", "raisin", "tobacco", "nut", "malty", "orange", "dried fruit", and "syrup". There are, however, specific differ-

ences within the cluster. For example, the frequency of describing "dark lager" with terms such as "orange", "dried fruit", or "syrup" is higher than with "porter and stout". These differences can also be related to the sensory differences of the beer styles classified within the same group. Therefore, the next step is to characterize the sensory profile base of the beer styles.

Subsequently, ale and lager data were analyzed separately. To avoid not including specific attributes which might contribute to the uniqueness of specific beer styles, new cut-off values were established. The criteria to establish the new cut-off values were based on the number of beers from the beer styles with a lower representation within ales and lagers, respectively (six citations for "Old ale" style in ale beers and eight for "kellerbier/zwickel" for lager).

Ale Beers

The biplot in Figure 5A illustrates the sensory space from the different ale beer styles. The first two dimensions (F1 and F2) account for 78.89% of the total explained variance (Figure 5A). The corresponding HCA (represented by circles in Figure 5A) showed the formation of three different clusters. Along F1 (65.94%), we found the largest cluster, which includes beers from Indian Pale Ale ("IPA"), American Pale Ale ("APA"), and "Double IPA" characterized by many different hop-related aroma attributes, from tropical fruit and floral descriptors to styles such as "Golden ale", "English ale", or "amber ale". The latter group is characterized by attributes such as "ginger", "dried fruit", or "cloves". The second cluster is represented by beer styles with a spicier character. The separation within these clusters is driven by F2 (12.95), "brown ale" and "IPA" being the largest contributors to the second dimension. The attributes "syrup" and "dried fruits" were the drivers for the separation along F2 (Figure 5). These results are in agreement with sensory studies such as the one performed by Jardim et al. [41], where IPA beers were characterized with "hoppy", "fruity", and "floral" aromas in comparison to other styles included in the study, such as "standard American lager" or "Irish red ale". Furthermore, the last clustered is located on the right side of F1. It is grouping styles from "British-American" such as "black IPA", with the different Porter beer styles: "Imperial Porter and stout", "dry porter and stout", and "sweet porter and stout". These beers are described with aroma attributes such as "coffee", "smoke", or "vanilla," among many others (Figure 5A).

Lager Beers

A similar procedure was followed for lager beers. Results are displayed on the biplot from the CA analysis (Figure 5B). The total explained variance for the first two dimensions accounts is 74.67%. In this case, most beer styles clustered together, except for "kellerbier/zwickel" and "Imperial and Indian pale lager", which formed two individual clusters. The style "Imperial and Indian pale lager" is contributing to the separation along F1 (60.58%). When looking at the attribute's contribution to F1, "apple" and "pineapple" are the significant drivers, followed by the term "hop". The style "stronger lager", "modern style", and "kellerbier/zwickel" are the major contributors for F2. The attributes "straw" and "dried fruit" are the major drivers for F2, and they are associated with "stronger lager" and "kellerbier/zwickel", respectively. The terms "lemon" and "butter" are driving the negative side of F2 (Figure 5B).

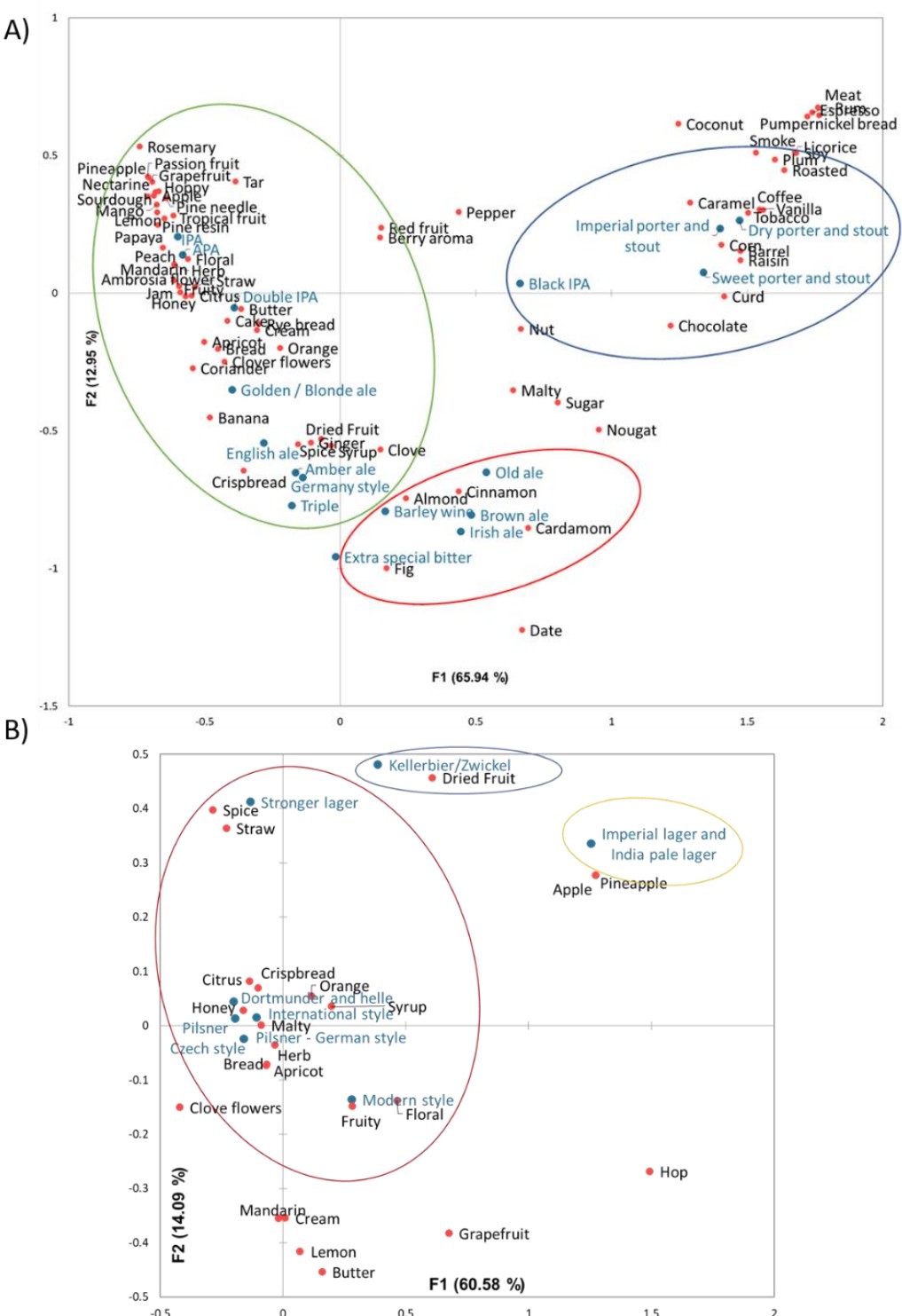

**Figure 5.** (**A**) Correspondence analysis (CA) biplot illustrating the distribution of the different ale styles based on the frequency of citation of sensory descriptors. (**B**) Correspondence analysis (CA) biplot illustrating the distribution of the different lager styles based on the frequency of citation of sensory descriptors. For each biplot, scores (colored in blue with blue dots) represent beer styles, whereas loadings (colored in black with red dots) represent sensory descriptors. Ellipses in different colours represent the clusters found with the corresponding hierarchical cluster analysis (HCA). IPA stands for Indian Pale Ale; APA stands for American Pale Ale.

### 3.3. Text Data as a Tool for Hop Characterization

Other studies have investigated the differences in hoppy beer flavors from a chemical or sensory perspective [5,42,43]. This section explores the potential usability of text data to characterize the sensory space of different raw materials, such as hop varieties. As previously mentioned, Systembolaget provides information about the different beer ingredients (i.e., hops). Focusing on extracted data for ale beers, we found 1583 mentions of hop-related information, 1009 being just an indication of the general term "humle" ("hop" in Swedish). Hop was mentioned more frequently in the IPA, APA, and Double IPA beer styles (288, 88, and 40 times respectively), with a total of 73 different hop cultivars being used. Nevertheless, the sensory characterization focused only on those hops used for at least 10% of these beers. The selected hop varieties for sensory characterization were the following: Amarillo, Cascade, Centennial, Chinook, Citra, Columbus, Galaxy, Magnum, Mosaic, Nelson Sauvin and Simcoe. Beers containing any other hop varieties were excluded.

Therefore, the sensory characterization was carried on 153 beers. Two binary tables were built based on the absence/presence of the following information: a binary table for sensory attributes, which was submitted to MCA (Multiple Correspondence Analysis) and another binary table with hops data included as a qualitative supplementary variable. Being included as supplementary data, the hop data did not influence the computations. However, it indicates how the hops are positioned on the correspondence map according to the sensory attributes. The categories were created on the absence/presence (0/1) base, for example, pineapple-0 (i.e., not used as a descriptor) and pineapple-1 (i.e., used as descriptors). Therefore, to facilitate visualization, labels from categories associated with absence (-0) were removed from the plot. The sensory outcomes are discussed and compared to the latest annual report from the global supplier of hop products, BarthHass GmbH & Co. KG (Nuremberg, Germany) [44].

In short, a map (Figure 6) is obtained projecting the coordinates of the sensory attributes (red with a yellow background) in the factors space. Projections of hop coordinates are displayed in black. To better interpret the results, the contributions, test values, and square cosines were also checked. Based on test-values, the following attributes are significant to the map ($p < 0.05$): "apple", "pineapple", "passion fruit", "grapefruit", "hop", "malty", "syrup", and "dried fruit". On the negative side of F1 (65.61%), we can observe "pineapple" and "apple" as the major contributors to the axis. The "passion fruit", "pine needles", or "mango" also related to the use of hops such as Mosaic or Citra and, Simcoe. On the positive side of F1, we can observe hops such as Amarillo, Chinook or Magnum with attributes such as "spicy", "dried fruits", or "malty". Magnum is traditionally used as a bittering hop. However, it also releases subtle "spicy" or "dried fruit" aromas [44]. Not being excessively aromatic may favor other aromas related to other ingredients (i.e., malt) or brewing steps. The second dimension, F2, is driven by the presence of "citrus" aromas on the characteristics of Columbus hops [44]. Apricot and orange have a more significant contribution to the negative side of F2 (6.31%), associated with Centennial hops. Nelson Sauvin and Galaxy are very close to 0 and therefore was more difficult to associate them with specific descriptors.

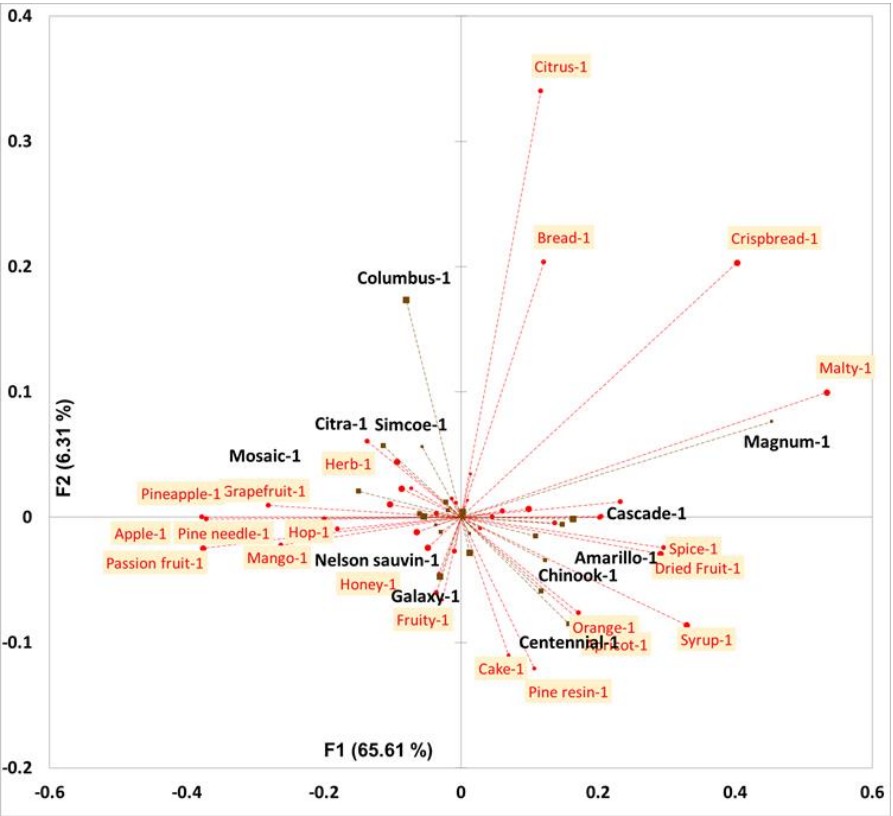

**Figure 6.** Map of categories projecting the coordinates from the sensory attributes (red) and the presence of hops as supplementary data (black). F1—first dimension, F2—second dimension.

## 4. Conclusions

The present work has provided an overview of the status of the Swedish beer market and the sensory space of its beers. Swedish market has a clear preference for hoppy beers, with a large number of IPA beers within the alcohol retail monopoly. Different data strategies (heatmaps, correspondence analysis or multiple correspondence analysis) have been used to characterize the sensory space of different Swedish beers types (ale and lager beers) and beer styles. Additionally, the multiple correspondence analysis has shown that text data can potentially characterize the sensory space of ingredients, in this case, hops. This work is an example to highlight the potential usability of text data as a cost-effective way to characterize, for example, the sensory space of specific products. Data scientists could apply different criteria or strategies within the same dataset, from trying to understand the role of hop combination related to specific aroma attributes to including the type of malt as an extra variable. It can help not only to understand the demographics of a specific market but to understand their sensory profile.

**Supplementary Materials:** The following are available online at https://www.mdpi.com/article/10.3390/beverages7040074/s1, Figure S1: Average beer price (SEK/L) per county, Figure S2: Pie-chart illustrating the representative % of the different beer styles from ale (A) and lager beers (B).

**Author Contributions:** Conceptualization: G.G.-B. and M.M.; Data curation: H.d.B.A. and G.G.-B.; Formal analysis; H.d.B.A. and G.G.-B.; Methodology: G.G.-B. and M.M.; Project administration; Resources; Software; Supervision; G.G.-B. and M.M.; Validation: G.G.-B., H.d.B.A., and M.M.; Visualization: G.G.-B.; Roles/Writing—original draft; G.G.-B., H.d.B.A.; Writing—review & editing: G.G.-B., H.d.B.A., and M.M. All authors have read and agreed to the published version of the manuscript.

**Funding:** This research part of Hope for Swedish hops (EIP-AGRI, Agriculture & Innovation), funded by Jordbruksverket 2018-2132.

**Institutional Review Board Statement:** Not applicable.

**Informed Consent Statement:** Not applicable.

**Data Availability Statement:** Not applicable.

**Conflicts of Interest:** The authors declare no conflict of interest.

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
