# Peer review of "Mapping the Sensory Fingerprint of Swedish Beer Market through Text and Data Mining and Multivariate Strategies"

_beverages, doi:10.3390/beverages7040074_

Round 1

Reviewer 1 Report

Journal: Beverages

Title: Mapping the sensory fingerprint of Swedish beer market through text and data mining and multivariate strategies

Authors: Gonzalo Garrido-Bañuelos*, Helia de Barros Alves and Mihaela Mihnea

The sensory space of Swedish beers were investigated and determined. The idea is sound and striking, the realization is reasonable.

I would bound “sensory morphological wheel” to Magni Martens [Magni Martens*, Siren R. Veflingstad, Erik Plahte, Dominique Bertrand e, Harald Martens: A sensory scientific approach to visual pattern recognition of complex biological systems. Food Quality and Preference 21 (2010) 977–986. doi:10.1016/j.foodqual.2010.04.013

Similarly, I would also cite such papers as [Dániel Koren, Laura LÅ‘rincz, Sándor Kovács, Gabriella Kun-Farkas, Beáta Vecseriné Hegyes, László Sipos: Comparison of supervised learning statistical methods for classifying commercial beers and identifying patterns. Journal of Chemometrics. 34 (2020) e3216. DOI: 10.1002/cem.3216 ]

For same reason I gathered a paper about antioxidant activities of commercial beers, which is an important aspect, but it seems to have nothing to do with the present investigation: [Haifeng Zhao*, Wenfen Chen, Jian Lu, Mouming Zhao: Phenolic profiles and antioxidant activities of commercial beers Food Chem. 119 (2010) 1150-1158 doi:10.1016/j.foodchem.2009.08.028 ]

Minor errors

“The present aims to overview” – i) What? (study), ii) “The present [study] aims to overview…”” – the formulation is unfortunate. It means, because of the English understatement, that the aims could not be reached. The authors’ message contradicts this.

“The biplot in Figure 5A” – Figure 5 is not a biplot, for a biplot Figures 5 and 6 should have been united.

“CA or MCA” – abbreviations should not be used in titles, abstract and conlcusions.

November 14 / 2021               referee:

Author Response

Comments and Suggestions for Authors

Journal: Beverages

Title: Mapping the sensory fingerprint of Swedish beer market through text and data mining and multivariate strategies

Authors: Gonzalo Garrido-Bañuelos*, Helia de Barros Alves and Mihaela Mihnea

The sensory space of Swedish beers were investigated and determined. The idea is sound and striking, the realization is reasonable.

I would bound “sensory morphological wheel” to Magni Martens [Magni Martens*, Siren R. Veflingstad, Erik Plahte, Dominique Bertrand e, Harald Martens: A sensory scientific approach to visual pattern recognition of complex biological systems. Food Quality and Preference 21 (2010) 977–986. doi:10.1016/j.foodqual.2010.04.013

Similarly, I would also cite such papers as [Dániel Koren, Laura LÅ‘rincz, Sándor Kovács, Gabriella Kun-Farkas, Beáta Vecseriné Hegyes, László Sipos: Comparison of supervised learning statistical methods for classifying commercial beers and identifying patterns. Journal of Chemometrics. 34 (2020) e3216. DOI: 10.1002/cem.3216 ]

For same reason I gathered a paper about antioxidant activities of commercial beers, which is an important aspect, but it seems to have nothing to do with the present investigation: [Haifeng Zhao*, Wenfen Chen, Jian Lu, Mouming Zhao: Phenolic profiles and antioxidant activities of commercial beers Food Chem. 119 (2010) 1150-1158 doi:10.1016/j.foodchem.2009.08.028 ] 

The authors considered the suggested references to enrich que quality of the paper. As the work was performed in beers, the reference (Koren et al. 2020 Comparison of supervised learning statistical methods for classifying commercial beers and identifying patterns. Journal of Chemometrics) was integrated within the text:

Line 57-58: Different statistical methods have also been explored in beers [32].

Other references have also been added: 

Mafata, M. PhD Thesis. Stellenbosch University. A chemometric approach to investigating South African wine behaviour using chemical and sensory markers, 2021

Hopfer, H.; McDowell, E.H.; Nielsen, L.E.; Hayes, J.E. Preferred beer styles influence both perceptual maps and semantic descriptions of dry hops. Food Qual. Prefer. 2021, 94, 104337, doi:10.1016/j.foodqual.2021.104337.

Kankolongo Cibaka, M.L.; Ferreira, C.S.; Decourrière, L.; Lorenzo-Alonso, C.J.; Bodart, E.; Collin, S. Dry hopping with the dual-purpose varieties Amarillo, Citra, Hallertau Blanc, Mosaic, and Sorachi ace: Minor contribution of hop terpenol glucosides to beer flavors. J. Am. Soc. Brew. Chem. 2017, 75, 122–129, doi:10.1094/ASBCJ-2017-2257-01

Minor errors

“The present aims to overview” – i) What? (study), ii) “The present [study] aims to overview…”” – the formulation is unfortunate. It means, because of the English understatement, that the aims could not be reached. The authors’ message contradicts this.

The word “work” was added.

“The biplot in Figure 5A” – Figure 5 is not a biplot, for a biplot Figures 5 and 6 should have been united.

The following text was added to the Figure caption to clarify:

For each biplot, scores (colored in blue with blue dots) represent beer styles, whereas loadings (colored in black with red dots) represent sensory descriptors. Ellipses represent the clusters found with the corresponding Hierarchical Cluster Analysis (HCA).

“CA or MCA” – abbreviations should not be used in titles, abstract and conlcusions.

Changed accordingly

Reviewer 2 Report

Dear authors,  this work was interesting but I would like to suggest some modifications and corrections to increase the quality of the paper.

Line 55: please add a dot after “et al”

Line 85: many parts of fig 1 are impossible to read. I would suggest to increase the quality of the image or to replace / modify this one. In general I would like to suggest improving the quality of all figures.

Line 86: please remove the second dot at the end of the sentence

Line 92: please remove doble space before “Fisher's least”

Line 78: please can you explain why you didn’t consider the “sour beers”

Line 80: please remove doble space before “name of beer”

Line 147: please remove doble space before “On many occasions”

Line 290: the supplementary material, which in the text is indicated as S1 and S2, in the folder has the name of Figure 1 and Figure 2. Please correct.

I would strongly encourage the citation of more recent articles considering that half of the bibliography cited is prior to 2015. In this regards, I suggest citing the following article:

Brewing Quality of Hop Varieties Cultivated in Central Italy Based on Multivolatile Fingerprinting and Bitter Acid Content. Mozzon M., Foligni R., Mannozzi C. Foods. 9(5), foods9050541 (2020). DOI: 10.3390/foods9050541

Author Response

 Comments and Suggestions for Authors

Dear authors,  this work was interesting but I would like to suggest some modifications and corrections to increase the quality of the paper.

Line 55: please add a dot after “et al”

Added

Line 85: many parts of fig 1 are impossible to read. I would suggest to increase the quality of the image or to replace / modify this one. In general I would like to suggest improving the quality of all figures.

Figure 1 has been slightly enlarged and the rest of figures have been pasted with higher quality.  

Line 86: please remove the second dot at the end of the sentence

Removed

Line 92: please remove doble space before “Fisher's least”

Removed

Line 78: please can you explain why you didn’t consider the “sour beers”

It was simply a matter of choice. The authors decided to not include “sour beers” as they were a reduced number, and they did not have any subcategories (beer type of styles).

Line 80: please remove doble space before “name of beer”

Removed

Line 147: please remove doble space before “On many occasions”

Removed

Line 290: the supplementary material, which in the text is indicated as S1 and S2, in the folder has the name of Figure 1 and Figure 2. Please correct.

Adjusted within the text and Supplementary file

I would strongly encourage the citation of more recent articles considering that half of the bibliography cited is prior to 2015. In this regards, I suggest citing the following article:

Brewing Quality of Hop Varieties Cultivated in Central Italy Based on Multivolatile Fingerprinting and Bitter Acid Content. Mozzon M., Foligni R., Mannozzi C. Foods. 9(5), foods9050541 (2020). DOI: 10.3390/foods9050541

Added

Other recent works were also added as references :

In line 56-57: “Mafata investigated different data fusion strategies to understand the relationship between chemical and sensory markers in wine [31]”.

Reference: Mafata, M. PhD Thesis. Stellenbosch University. A chemometric approach to investigating South African wine behaviour using chemical and sensory markers, 2021

Other studies have investigated the differences in hoppy beer flavors from a chemical or sensory perspective [5,42,43].

Hopfer, H.; McDowell, E.H.; Nielsen, L.E.; Hayes, J.E. Preferred beer styles influence both perceptual maps and semantic descriptions of dry hops. Food Qual. Prefer. 2021, 94, 104337, doi:10.1016/j.foodqual.2021.104337.

Kankolongo Cibaka, M.L.; Ferreira, C.S.; Decourrière, L.; Lorenzo-Alonso, C.J.; Bodart, E.; Collin, S. Dry hopping with the dual-purpose varieties Amarillo, Citra, Hallertau Blanc, Mosaic, and Sorachi ace: Minor contribution of hop terpenol glucosides to beer flavors. J. Am. Soc. Brew. Chem. 2017, 75, 122–129, doi:10.1094/ASBCJ-2017-2257-01.